# An Empirical Study of Multimodal Model Merging

**Yi-Lin Sung[1], Linjie Li[2], Kevin Lin[2], Zhe Gan[3][*], Mohit Bansal[1], Lijuan Wang[2]**
[1]UNC Chapel Hill, [2]Microsoft, [3]Apple AI/ML
{ylsung, mbansal}@cs.unc.edu
{Lindsey.Li, keli, lijuanw}@microsoft.com
zhe.gan@apple.com

## Abstract

Model merging (*e.g.*, via interpolation or task arithmetic) fuses multiple models trained on different tasks to generate a multi-task solution. The technique has been proven successful in previous studies, where the models are trained on similar tasks and with the same initialization. In this paper, we expand on this concept to a multimodal setup by merging transformers trained on different modalities. Furthermore, we conduct our study for a novel goal where we can merge vision, language, and cross-modal transformers of a modality-specific architecture to create a parameter-efficient modality-agnostic architecture. Through comprehensive experiments, we systematically investigate the key factors impacting model performance after merging, including initialization, merging mechanisms, and model architectures. We also propose two metrics that assess the distance between weights to be merged and can serve as an indicator of the merging outcomes. Our analysis leads to an effective training recipe for matching the performance of the modality-agnostic baseline (*i.e.*, pre-trained from scratch) via model merging. Our method also outperforms naive merging significantly on various tasks, with improvements of 3% on VQA, 7% on COCO retrieval, 25% on NLVR$^2$, 14% on Flickr30k and 3% on ADE20k.[1]

## 1 Introduction

Model merging (Singh and Jaggi, 2020; Matena and Raffel, 2022; Ainsworth et al., 2022; Jordan et al., 2022), investigates the technique of merging (*e.g.*, via linear interpolation, task arithmetic (Ilharco et al., 2022), RegMean (Jin et al., 2023)) two models trained on different tasks while preserving their original capabilities or even outperforming multi-task training (Li et al., 2022). This technique

enables us to generate the multi-task solution without synchronous training and is especially useful when the models for merging are already trained and available online. However, current literature has only applied model merging to models trained on similar tasks or even the same dataset, which inspires us to explore whether model merging will also work when the models are trained on different modalities such as language versus vision.

We study this challenging problem with a novel goal of obtaining an effective and parameter-efficient modality-agnostic model by merging the different modality transformers of a modality-specific model. In vision-language (VL) domains (Driess et al., 2023; OpenAI, 2023), many modality-specific models (Tan and Bansal, 2019; Chen et al., 2020; Radford et al., 2021; Li et al., 2021; Dou et al., 2022) employ dedicated transformer encoders to encode vision/language inputs independently, and on top of which additional transformer layers are used for multimodal fusion. On the other hand, the convergence of using transformer-based architectures for various single-modality tasks has prompted researchers to adopt a single modality-agnostic transformer (Akbari et al., 2021; Wang et al., 2021; Kim et al., 2021; Lu et al., 2022; Jaegle et al., 2022) to encode different modality inputs and learn cross-modal interactions simultaneously. While modality-specific architectures usually perform better, modality-agnostic ones are simpler and more parameter-efficient. Both architecture designs offer their unique benefits and have distinct use cases. However, obtaining the benefits of both these setups can be costly as it requires independent training. Therefore, in this paper, we explore how to leverage a well-trained modality-specific model and develop a modality-agnostic model from it via model merging, with a goal of achieving similar performance to the modality-agnostic baseline which is pre-trained

---

[*]Work done at Microsoft

[1]Our code is available at https://github.com/ylsung/vl-merging

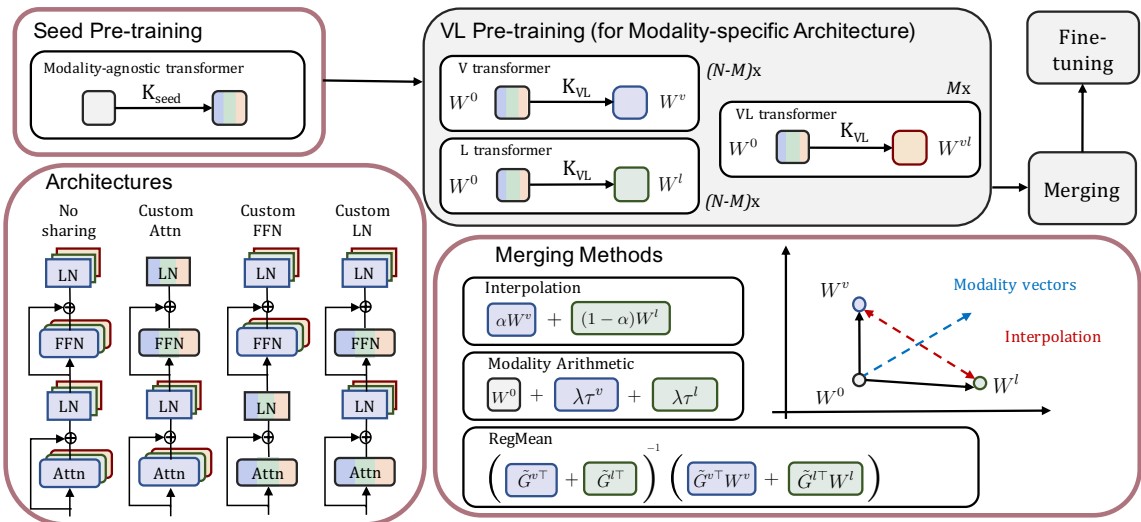

Figure 1: Overview of our framework for **multimodal merging**. The boxes with red boundaries illustrate the factors (seed pre-training, merging methods, architectures) that are crucial for the performance of merging. We use blue-filled, green-filled, and orange-filled colors to denote vision, language, and VL modalities, respectively.

from scratch.[2] This approach eliminates the need for training the latter architecture.

To this end, we conduct a systematic study of model merging on VL models, analyzing the key factors impacting the merging results, including initialization, merging mechanisms, and model architectures. Through comprehensive experiments, we find that initialization is crucial for model merging, and using pre-trained weights on VL data is helpful. We benchmark three merging methods: interpolation, modality arithmetic, and RegMean, and we find that interpolation performs competitively while being simple and efficient among compared merging methods. Lastly, we examine the merging of different modality-specific architecture variants, including shared-weight architectures with custom attention layers, feed-forward networks (FFN), or layer normalization layers. We find that the architecture without sharing performs the best before merging and attains on-par performance to the modality-agnostic baseline after merging.

Moreover, we investigate the correlation between the distance of two weights to be merged and the merging results, and propose two distance measures that can potentially indicate the merging results before training. Overall, our findings serve as a training recipe which can significantly improve the simple merging of the modality-specific model (by around 3% on VQA, 7% on COCO image-text retrieval, 25% on NLVR[2], 14% on Flickr30k and

3% on ADE20k), to match the performance of the modality-agnostic model obtained by independent training, demonstrating a novel way to construct a universal architecture that can process different modalities simultaneously. We also demonstrate the merged weight maintains the performance on unimodal tasks (we focus on vision tasks), and the proposed approach generalizes broadly to another backbone model (ViT-Tiny/16).

## 2 Related Work

**Different Types of VL Architectures.** VL models aim to generate outputs based on reasoning over both text and visual inputs. Because the model takes two sources of inputs, there are different design choices to fuse the information, and hence yielding several mainstream designs in VL modeling: (1) modality-specific transformer (Tan and Bansal, 2019; Li et al., 2023; Driess et al., 2023), (2) modality-agnostic transformer (Akbari et al., 2021; Wang et al., 2021; Kim et al., 2021; Aghajanyan et al., 2022), and (3) dual transformer (Radford et al., 2021). Modality-specific transformers use a language transformer and a visual encoder to extract modality-specific representations and feed them into a cross-modal transformer to learn image-text fusion information. Some modern architectures (Driess et al., 2023; Sung et al., 2022b; Cho et al., 2021; Yu et al., 2022; Wang et al., 2022b; Sung et al., 2022a) use the language transformer for both language and cross-modal modeling. The dual transformer is a special case of modality-specific architecture, where the cross-modal fusion is realized

---

[2]When using model merging, we sometimes refer to the modality-specific model as the model *before merging*, and the modality-agnostic one as the model *after merging*.

via simple dot product, hence more efficient for fast retrieval. Modality-agnostic transformers use one single transformer to process multi-modal data. These methods can be viewed as early fusion models, and the transformer learns modality-specific and cross-modal information simultaneously. The design of modality-agnostic architectures is relatively simple, and they are more memory-efficient because they only have one backbone.

**Model Merging.** The *de facto* approach in recent model development centers around the "pre-train-then-fine-tune" paradigm. Due to the substantial resources required to train robust pre-trained models, the available repertoire is relatively limited, and many subsequent models are fine-tuned from this finite selection. Recent research has discovered that models derived from the same pre-trained weights can be successfully merged into a single model. This merge can either improve single-task performance when the models are designed for the same task (Wortsman et al., 2022a) or create a multi-task solution when they target different tasks (Ilharco et al., 2022; Wortsman et al., 2022b; Don-Yehiya et al., 2022). Various strategies for merging models have been proposed, including linear interpolation, task arithmetic (Ilharco et al., 2022), and RegMean (Jin et al., 2023). However, if models do not share the same pre-trained weights, weight permutation might be required before merging (Entezari et al., 2021; Ainsworth et al., 2022; Jordan et al., 2022). Our work leverages model merging towards the innovative goal of generating an efficient modality-agnostic model from modality-specific models, thus eliminating the need for re-training.

## 3 VL Transformers

A VL model takes an image-text pair as input and is designed to tackle various VL tasks with minimal architecture changes. The input image is first divided into patches, flattened into a sequence, and then fed into the vision embedding layer to obtain visual features. The input text is processed by the language embedding layer to obtain a sequence of textual features. The features are then jointly processed by the VL model, for which we explore different architectures detailed below.

**Modality-specific transformer** contains two $N$-layer ($N$=12 in this paper) transformers, specialized for encoding image and text inputs, respectively, and an additional $M$ (=2 in this paper) transformer layers for multimodal fusion (Akbari et al.,

2021; Lu et al., 2019; Li et al., 2019; Su et al., 2020). We denote their weights for layer $i$ as $W_i^v$, $W_i^l$ ($i = 1, 2, ..., 12$) and $W_i^{vl}$ ($i = 11, 12$). Please refer to Figure 2(a) for illustration. This design is inspired by VLMo (Wang et al., 2022a, 2023), which has different routes for different tasks in forwarding. For visual question answering and other classification tasks, we use the first $(N - M)$ layers of vision and language transformers to extract the image and text features and feed the concatenation of them to the $M$ multimodal fusion layers to get the final representation. We demonstrate this architecture/mechanism in the upper part of Figure 1 and the right part of Figure 2(b). For image-text retrieval, we directly use the whole $N$ layers of vision and language transformers to extract image and text representations for matching (left part of Figure 2(b)). Only using unimodal encoders to extract features allows us to compute the dot product between features efficiently because it eliminates the need to forward the input pairs to cross-modal layers. Please refer to VLMo for more details. For easier merging, we initialize all the transformers from the same initialization[3], regardless of the modalities they are used for, and we use $W_i^0$ to denote the initial weight for the $i^{th}$ layer.

**Modality-agnostic transformer** adopts one set of parameters to jointly learn vision, language, and cross-modal information, that is, $W_i^v = W_i^l$ for $i = 1, 2, ..., 12$ and $W_i^v = W_i^l = W_i^{vl}$ for $i = 11$ and 12. The forward mechanism of this model is the same as the modality-specific architecture.

**Shared-weight modality-specific transformer** is in-between the two aforementioned extreme types of architecture, *i.e.*, with partial modality-agnostic and modality-specific modules. Specifically, we explore three variants of the shared-weight architectures with: (1) custom attention layers, (2) custom FFN, and (3) custom layer normalization layers. The illustration of them is shown in Figure 1.

## 4 Model Merging for VL Architectures

Our objective is to combine various transformers within the modality-specific architectures into a single modality-agnostic transformer. We do not merge the embedding layers and keep them modality-specific as they have different architec-

---

[3]We have explored merging the model that uses BERT to initialize the language transformer and BEiT for the other transformers, but the average of IR and TR is only 49.9 on COCO even if we permute the weights (80.6 before merging).

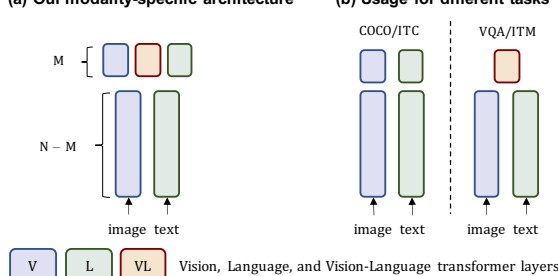

**(a) Our modality-specific architecture**     **(b) Usage for different tasks**

COCO/ITC     VQA/ITM

M

N − M

image text     image text     image text

V    L    VL    Vision, Language, and Vision-Language transformer layers

Figure 2: VLMo-like modality-specific architecture. It has an N-layer vision, N-layer language, and M-layer cross-modal transformer. For classification tasks and ITM in pre-training, it uses cross-modal layers to jointly process the image and text input. For retrieval tasks and ITC in pre-training, it only forwards inputs through vision and language encoders. For modality-agnostic architecture, the V, L, and VL transformer layers share a single set of parameters while keeping the forward mechanism the same.

tures and dimensions. For shared-weight architectures, we only merge the weights of custom layers since the other parts are already modality-agnostic. In a modality-specific VL model, the transformers receive inputs from different modalities and work together to accomplish the target task. We merge such transformers to allow a single transformer that can handle different modalities concurrently. This situation deviates from previous studies on model merging (Li et al., 2022; Matena and Raffel, 2022; Ainsworth et al., 2022), where the transformers (in these previous works) to be merged take input from the same modality. In the following, we describe our process for merging the VL architectures.

### 4.1 Merging Methods

We explain the details of the three merging methods, and their illustrations are shown in Figure 1.

**Interpolation.** For simple interpolation, the vision, language, and cross-modal transformer layers are element-wise weighted-averaged with given ratios. We control the ratio $\alpha$ between the vision and language transformer and set the ratio of the cross-modal transformer at a constant value ($\frac{1}{3}$ for the layers that have three modalities). The merged weights are $\alpha W_i^v + (1-\alpha) W_i^l$ for the first $(N-M)$ layers while they are $\frac{2}{3}(\alpha W_i^v + (1-\alpha) W_i^l) + \frac{1}{3} W_i^{vl}$ for the top $M$ layers.

**Modality Arithmetic.** The second method is inspired by the concept of task vectors (Ilharco et al., 2022), which indicate the directions to improve the tasks' performance from the initial weight. For a given task, the task vector is ob-

tained by subtracting the initial weight from the tuned weight (illustrated in Figure 1). Then, the merged weight containing multi-task information is computed by adding all task vectors to the initial weight. In this paper, we extend this idea to learning such vectors for different modalities, dubbed as modality vectors. We define the modality vectors as the difference between the weight for a specific modality and the initial weight, for example, $\tau_i^v = W_i^v - W_i^0$, and the merged weight is $W_i^0 + \lambda \sum_{m \in \{v,l\}} \tau_i^m$ for the first $(N-M)$ layers and is $W_i^0 + \lambda \sum_{m \in \{v,l,vl\}} \tau_i^m$ for the top $M$ layers, and $\lambda$ is the scaling parameter.

**RegMean.** Different from the previous two methods, RegMean (Jin et al., 2023) finds closed-form solutions for the weights of linear layers and evenly interpolates other weights (layer normalization, bias terms). Assume $X_i^m$ is the input for the $i^{th}$ layer in the transformer for $m$ modality, RegMean finds the merged weight $W_i$ that minimizes the distance of features before and after merging, that is, $\sum_{m \in \{v,l,vl\}} \|W_i^\top X_i^m - W_i^{m\top} X_i^m\|_2^2$. The closed-form solution of $W_i$ is $(\sum_{m \in \{v,l,vl\}} G_i^m)^{-1} \sum_{m \in \{v,l,vl\}} (G_i^m W_i^m)$, where $G_i^m = X_i^{m\top} X_i^m$. Moreover, Jin et al. (2023) pointed out that introducing a scalar $\gamma$ to decrease the non-diagonal elements of $G_i^m$ ($\tilde{G}_i^m = \gamma G_i^m + (1-\gamma)\text{diag}(G_i^m)$) and replace $G_i^m$ by $\tilde{G}_i^m$ in the above solution form leading to stabler results.

### 4.2 Evaluation Pipeline

Unlike previous model merging literature that merges models fine-tuned on downstream tasks,[4] we merge the transformer weights for different modalities after VL pre-training (Gan et al., 2022) to explore a more effective modality-agnostic pre-trained model for fine-tuning. We follow the same pipeline to evaluate different merging methods as shown in the upper part (gray-filled boxes) of Figure 1. First, we pre-train the modality-specific models for $K_{VL}$ steps on VL corpra in an end-to-end manner. Next, we merge the weights of modality-specific transformers to construct a modality-agnostic model via the methods described in Section 4.1. Lastly, we fine-tune the merged model on downstream tasks. Note that when reporting the performance of modality-specific architectures, weight merging is not needed. Hence, we

---

[4]We find that directly merging a fine-tuned modality-specific model performs poorly: the average of IR and TR is 50.5 on COCO (80.5 without merging).

| Method | VQA | MSCOCO | |
| --- | --- | --- | --- |
| | Acc | Avg TR | Avg IR |
| Modality-agnostic architecture (MAA) VL pre-trained from scratch | 73.94 | 83.91 | 72.23 |
| Modality-specific architecture (MSA) VL pre-trained from scratch | 75.14 | 86.46 | 74.67 |
| Merging from MSA by interpolation (merging baseline) | 70.89 | 77.38 | 65.43 |

Table 1: Results on directly merging the modality-specific architecture, without seed pre-training, and the comparison with the modality-agnostic baseline. The merging result without seed pre-training has a significant performance gap to the other approaches.

directly finetune the models after pre-training.

We pre-train our models on the standard 10M image-text corpus from COCO (Lin et al., 2014), Visual Genome (Jin et al., 2022), SBU (Ordonez et al., 2011), and Conceptual Captions (Sharma et al., 2018) with three popular objectives: image-text contrastive (ITC) learning (Radford et al., 2021), image-text matching (ITM) (Li et al., 2021), and masked language modeling (MLM) (Devlin et al., 2019). During finetuning, we evaluate several VL and vision downstream tasks. The details of the experimental setup are shown in Section 5.

### 4.3 Seed Pre-training

For our first pilot investigation, we follow the above evaluation pipeline to pre-train a modality-specific model initialized from BEiT (Bao et al., 2022) for 200k iterations, then perform merging with interpolation ($\alpha = 0.5$), and lastly fine-tune on downstream tasks. We name this simple method as the **merging baseline**. Table 1 shows that the performance drop from merging is quite significant, and the merged model lags behind our goal of matching the modality-agnostic baseline, which is pre-trained from scratch. We hypothesize that the model weights of language and cross-modal transformers after VL pre-training are not located in the same basin (Entezari et al., 2021) as the vision transformer, because the initial BEiT weight only contains visual information.

Hence, in order to ensure that different transformers remain situated within a common basin and avoid significant deviation from one another after VL pre-training, we next turn to use a single pre-trained weight containing vision, language, and cross-modal information for initialization, by pre-training a single modality-agnostic transformer (initialized from BEiT) on VL corpora for $K_{seed}$ steps. This approach is also used by Li et al. (2022)

and referred as the *seed phase*. We then use this weight to initialize different modality transformers of the modality-specific model before the VL pre-training step and evaluate the model performance as described in Section 4.2. For a fair comparison, we ensure that all compared methods go through the same number of total pre-training iterations on the same VL corpora, *i.e.*, $K_{seed} + K_{VL} = 200k$.

Note that VL pre-training can be conducted on different model architectures, including both modality-specific models and modality-agnostic models. Also, seed pre-training for a modality-specific model is essentially equivalent to VL pre-training on a modality-agnostic model. We use different terms for seed pre-training and VL pre-training to distantiate the different training stages for training the modality-specific model.

## 5 Experimental Details

We describe the experimental setup for our models and training in this section and more details about hyper-parameters for fine-tuning, computational usage, and the number of parameters of our models are included in Appendix A.

**Architectures.** We use ViT-B/16 (Dosovitskiy et al., 2021) (which has a similar architecture as BERT (Devlin et al., 2019)) throughout our experiments. The model has 12 transformer layers, and the dimension of the model is 768.

**VL Pre-training.** As described in Section 4.2, we adopt the standard 10M image-text corpus (4M images) and utilize ITC, ITM, and MLM for VL pre-training. The overall pre-training objective is the weighted combination of the three aforementioned loss functions. ITC maximizes the pair-wise positive matching and minimizes the matching between any negative pairs within the training batch. Similar to ITC, ITM predicts whether the image-text pair is positive or negative, by modeling it as binary classification. For MLM, we mask out $15\%$ of text tokens, and the model is required to reconstruct the missing tokens with multimodal context. The overall pre-training objective is the weighted combination of the three aforementioned loss functions, and we find that using a smaller weight (0.25) for MLM empirically leads to better results. The forward mechanism of ITC is the same as the retrieval task, and of ITM and MLM is the same as VQA (Figure 2(b)).

The image input size is set as $224 \times 224$ and the

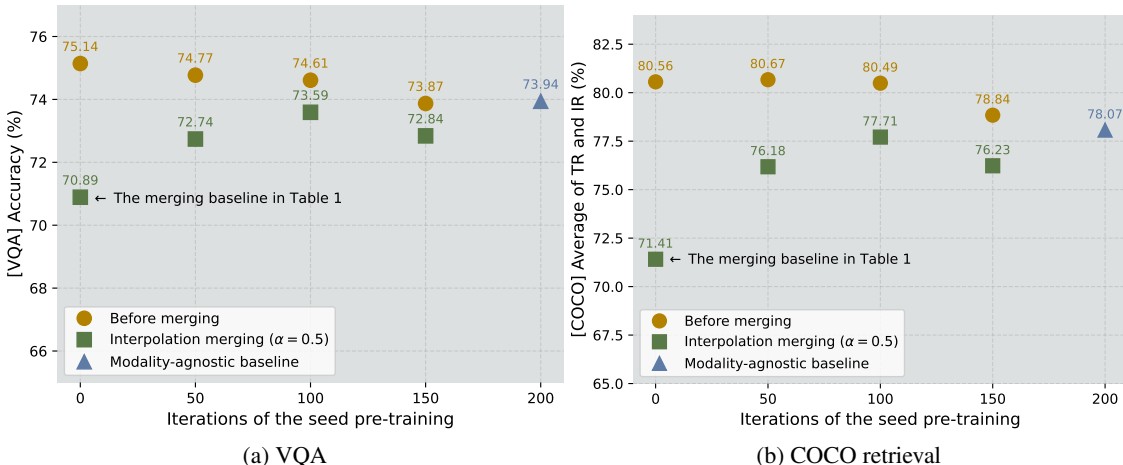

(a) VQA

(b) COCO retrieval

Figure 3: The effect of seed pre-training's duration on the merging results. Increasing the iterations of seed pre-training improves the merging outcomes, and the optimality happens at 100k iterations as more iterations in seed pre-training reduce the iterations of VL pre-training, and limit the learning of the modality-specific model.

| (a) Interpolation | | | | (b) Modality arithmetic | | | | (c) RegMean | | | |
|---|---|---|---|---|---|---|---|---|---|---|---|
| $\alpha$ for interpolation | VQA | COCO | | $\lambda$ for modality arithmetic | VQA | COCO | | $\gamma$ for RegMean | VQA | COCO | |
| | Acc | Avg TR | Avg IR | | Acc | Avg TR | Avg IR | | Acc | Avg TR | Avg IR |
| 0 (L weight only) | 68.58 | 75.25 | 62.64 | 0 (no VL PT) | 73.44 | 83.27 | 70.84 | 0 | 74.14 | 83.73 | 72.03 |
| 0.25 | 72.40 | 81.34 | 69.29 | 0.25 | **73.85** | 83.56 | 71.61 | 0.25 | **74.17** | 83.64 | **72.20** |
| 0.5 | 73.59 | 83.64 | 71.77 | 0.5 | 73.59 | **83.64** | **71.77** | 0.5 | 74.13 | **83.80** | 72.18 |
| 0.75 | **73.91** | **83.83** | **72.30** | 0.75 | 72.49 | 80.22 | 68.51 | 0.75 | 74.07 | **83.80** | 72.18 |
| 1.0 (V weight only) | 73.15 | 82.75 | 71.25 | 1.0 | 69.49 | 73.94 | 62.07 | 1.0 | 74.15 | 83.71 | 72.25 |

Table 2: The ablation of using different hyper-parameters for (a) interpolation, (b) modality arithmetic, and (c) RegMean. Note that interpolation with $\alpha = 0.5$ is the same as modality arithmetic with $\lambda = 0.5$. L, V, and PT are the abbreviations of language, vision, and pre-training. Interpolation works well when $\alpha = 0.5$ and $0.75$. With additional computations, RegMean can obtain stable performance.

text input length is 40. The text input is encoded by bert-base-uncased (Devlin et al., 2019) tokenizer, which has 30522 vocabularies. During training, we apply RandAugment (Cubuk et al., 2020) on image inputs. For all pre-training experiments, we set the batch size as 1024, the total number of iterations as 200k (if no seed pre-training), with 2500 for warmup. Adam optimizer with the peak learning 2e-4 is adopted for training.

**Fine-tuning.** We use VQA (Goyal et al., 2016), COCO image-text retrieval (Lin et al., 2014), NLVR$^2$ (visual reasoning) (Suhr et al., 2019), Flickr30k (image retrieval) (Plummer et al., 2015), ImageNet-1k (image classification) (Deng et al., 2009) and ADE20k (semantic segmentation) (Zhou et al., 2016) as downstream tasks. In evaluation, we report the accuracy for VQA and NLVR$^2$, top-1 accuracy for ImageNet, and mIOU for ADE20k. We use the average of top-1, top-5, and top-10 text recall (TR), and image recall (IR) for retrieval tasks. We report the results for each method with a single run. Appendix A shows other details of experiments.

## 6 Experimental Results

In this section, we investigate the key factors impacting the merging performance on VQA and COCO. Recall that our goal is to improve the merging baseline in Table 1 to match the performance of the modality-agnostic model and keep the performance of the modality-specific model (before merging) to remain similar.

### 6.1 Effect of Seed Pre-training

We first validate the effectiveness of seed pre-training by varying $K_{seed}$ in Figures 3a and 3b.[5] Training more steps in the seed phase naturally leads to worse performance for modality-specific models as fewer training steps are dedicated to the learning of modality-specific information in custom modules. However, **the merging performance is stronger for longer seed pre-training**, as the initial weight may absorb more language and cross-modal information, making the weights more likely to locate in the same basin before merg-

---

[5]Note that the models with $K_{seed} = 0$ is the same as what have been reported in Table 1.

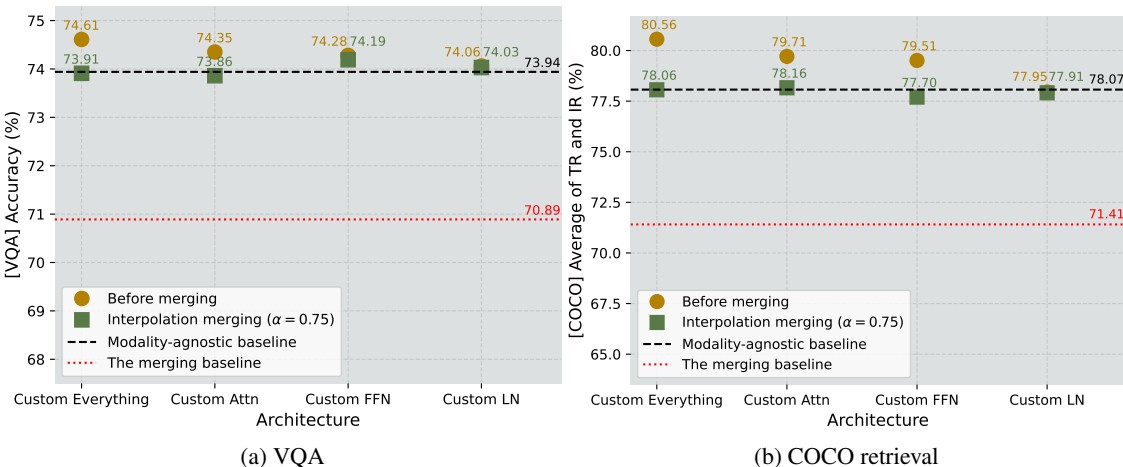

|   (a) VQA   |   (b) COCO retrieval   |

Figure 4: The effect of shared-weight architectures on the merging results. The modality-specific architecture which has entire custom modules performs the best before merging and attains on-par performance with the modality-agnostic baseline (pre-trained from scratch) after merging.

ing. Moreover, the optimality for merging happens when $K_{seed} = K_{VL} = 100k$, which also leads to a competitive modality-specific model that performs comparably to the one without seed pre-training. Therefore, we adopt this as the default recipe for all remaining experiments.

## 6.2 Comparison between Merging Methods

We compare different merging methods and ablate their hyper-parameters in Table 2. Table 2c shows that RegMean performs strongly in both tasks (especially VQA) and is robust to $\gamma$. Table 2b shows that reducing the length of the modality arithmetic leads to better downstream performance. In Table 2a, we observe that vision weight is more important for our downstream tasks, as only using vision weight ($\alpha = 1$) achieves much better results compared to only using language weight ($\alpha = 0$). Comparing different merging methods, **interpolation (with the ratio $0.75$) performs as competitively as RegMean without additional computation and storage for the gram matrices** $G_i^m$, and is more effective than modality arithmetic. Intuitively, the best performance with an interpolation ratio of $0.75$ is somewhat similar to using different weights for tasks to mitigate the negative transfer in multi-task learning (Liu et al., 2019).

## 6.3 Ablation on Shared-weight Architectures

Lastly, we explore applying model merging on different architectures described in Section 3, including shared-weight architectures with custom attention layers, FFN, or layer normalization layers. Results are shown in Figures 4a and 4b. The model with custom layer normalization layers can

largely retain the performance after merging, however its performance before merging is not as competitive, only comparable to the modality-agnostic baseline (whereas ideally, we want the performance of the modality-specific model before merging to be higher than the modality-agnostic model). Sharing attention layers or FFN has mixed results on VQA and COCO retrieval, as it improves on one task while hurts the other. Notably, **the modality-specific architecture with entirely independent modules performs the best without merging and attains on-par performance with the modality-agnostic baseline after interpolation**.

## 6.4 Final Recipe

Overall, as an initial study on merging modality-specific models, our empirical results reveal a recommended takeaway recipe: (1) select a pre-trained weight that has been trained on all modalities, (2) use interpolation with a ratio around $0.75$ (or $0.5$) as an effective initial approach, or employ RegMean if additional computational resources are deemed acceptable, and (3) employing modality-specific architectures with entirely independent modules can attain satisfactory merging outcomes.

## 6.5 Generalization to Diverse Tasks and Architectures

In this section, we further apply our merged model to a diverse spectrum of tasks and architectures and compare our approach with the modality-Specific Architecture (MSA), Modality-Agnostic Architecture (MAA), and the Merging Baseline (MB) (mentioned in Table 1 and Section 4.3).
**More Tasks.** We include additional evaluation on

| Methods | NLVR$^2$ | Flickr30k | | ImageNet-1k | ADE20k |
|---|---|---|---|---|---|
| | Acc | Avg TR | Avg IR | Top-1 Acc | mIOU |
| MSA | **79.71** | **95.10** | **87.37** | **83.25** | **49.85** |
| MAA | 78.39 | 93.10 | 84.57 | 82.78 | 48.65 |
| MB | 52.55 | 77.63 | 68.62 | 82.90 | 46.80 |
| Ours | **77.54** | **91.53** | **82.64** | **83.00** | **50.14** |

Table 3: Comparison between our method, Modality-Specific Architecture (MSA), Modality-Agnostic Architecture (MAA), and Merging Baseline (MB) across NLVR$^2$, Flickr30k, ImageNet-1k, and ADE20k. The top results are highlighted in bold for both merged and non-merged model categories. Our method aligns closely with MAA for NLVR$^2$ and Flickr30k and surpasses it on ImageNet-1k and ADE20k.

| Methods | VQA | NLVR$^2$ | COCO | | Flickr30k | |
|---|---|---|---|---|---|---|
| | Acc | Acc | Avg TR | Avg IR | Avg TR | Avg IR |
| MSA | **65.36** | **68.42** | **71.75** | **59.98** | **80.73** | **70.14** |
| MAA | 62.25 | 64.90 | 63.17 | 52.08 | 72.50 | 62.27 |
| MB | 55.52 | 51.07 | 51.98 | 42.43 | 17.63 | 16.09 |
| Ours | **60.02** | **52.89**\* | **63.40** | **52.82** | **68.40** | **59.53** |

Table 4: Comparison between our approach and baselines using ViT-Tiny/16 on four VL tasks. Our method demonstrates stronger performance compared to the merging baseline. \*We found using an interpolation ratio of 0.5 can boost the performance to 60.94 and cut the gap to MAA significantly.

NLVR$^2$ and Flickr30k as well as ImageNet-1k and ADE20k to cover more VL and vision tasks. To make a direct comparison, we do not apply the intermediate fine-tuning (used in VLMo) for Flickr30k, ImageNet-1k, and ADE20k. For modality-specific architectures, we only fine-tune the vision encoder to the vision tasks. The results of our experiments are presented in Table 3. Notably, our approach, which involves seed pre-training and interpolation with a ratio of 0.75, **exhibits significant improvements over the naive merging approach in both NLVR$^2$ and Flickr30k tasks**. However, we have observed that the performance gap between our approach and the modality-agnostic architecture is somewhat larger for NLVR$^2$ and Flickr30k compared to the VQA and COCO tasks, where our results closely align with the baseline performance (Figures 4a and 4b). We hypothesize that the merging outcomes could be better if the domain shift between pre-trained and downstream tasks were reduced, considering that images of VQA and COCO are used for VL pre-training. In the results of ImageNet and ADE20k, **our merging approach not only matches but also surpasses the baselines. These results affirm that the merging process does not compromise the representational capacity of the individual modalities.**

**More Architectures.** We also conduct experiments on another backbone model, ViT-Tiny/16 (Dosovitskiy et al., 2021). Note that it is not trained by the BEiT unsupervised objective but is trained by the supervised classification objective. In this experiment, we pre-train the model for 100k steps instead of 200k steps, and for our merging approach, we use 50k steps for both seed pre-training and VL pre-training. We compare our approach to baselines on four VL datasets and display the results in Table 4.

**Our findings demonstrate that our technique significantly outperforms the merging baseline.** Furthermore, we observe a larger performance gap between our approach and the modality-agnostic baseline when employing the ViT-Tiny/16 backbone, and this phenomenon is aligned with past research (Ainsworth et al., 2022), which shows larger (wider) models exhibit greater resilience to performance drop when merging.

## 7 Analysis

In the preceding experiments, we highlight three factors that influence the outcomes of merging. This sparked a curiosity regarding the existence of potential metrics capable of predicting or explaining the quality of these merging results. One intuitive hypothesis is that the distance between the weights to be merged (vision and language weights in our case) might play an important role, as the performance drop caused by merging should be 0 if the distance between them is 0, and vice versa, the drop should be considerable if the distance is large (the weights are not in the same basin). To test the hypothesis, we apply various distance measures on the vision and language weights of modality-specific models and compute the correlation between the distance and the performance drop after merging to assess the influence on merging results.

**Metrics Definition.** Four metrics are chosen to measure the distance between vision and language weights of modality-specific models, including the L2 distance, cosine dissimilarity (equivalent to one minus cosine similarity), soft sign dissimilarity (SSD), and truncated soft sign dissimilarity (TSSD), where the last two are inspired by Yadav et al. (2023). To compute the distance, we first flatten all the weight matrices from vision and language transformers and concatenate them respectively to form vision and language weight vectors

$\mathbf{w}^v$ and $\mathbf{w}^l$. Note that we do not include weights that are not mergeable, such as embedding layers. For L2 and cosine dissimilarity, we follow the standard ways to compute them, namely, $\sqrt{\sum_i (\mathbf{w}_i^v - \mathbf{w}_i^l)^2}$ and $1 - \frac{\mathbf{w}^v \cdot \mathbf{w}^l}{\|\mathbf{w}^v\|\|\mathbf{w}^l\|}$.

Yadav et al. (2023) show that model merging might result in information loss due to interference from redundant parameter values and disagreement on the sign of given parameter's values across weights to be merged. Motivated by this, we design two metrics to reflect the disagreement between signs and the interference of redundant parameters. Unlike the method used by Yadav et al. (2023), soft sign dissimilarity (SSD) considers the two requirements in a soft manner, and the mathematical definition is as follows,

$$\text{SSD}(\mathbf{x}, \mathbf{y}) = 1 - \frac{1}{L} \sum_i \frac{|\mathbf{x}_i + \mathbf{y}_i|}{|\mathbf{x}_i| + |\mathbf{y}_i|}, \quad (1)$$

where $L \ (= \|\mathbf{x}\| = \|\mathbf{y}\|)$ is the size of $\mathbf{x}$ and $\mathbf{y}$. Equation (1) represents one minus the soft sign similarity. The similarity will be one if $\mathbf{x}_i$ and $\mathbf{y}_i$ have the same sign, which can quantify the agreement of the signs. In the case when the values have different signs, the similarity will be the cancel-out value divided by the magnitude: (1) if they have exact opposite values, the similarity will be zero (the numerator is zero); (2) when $\mathbf{y}_i$ is a redundant value, denoting it has the opposite sign of $\mathbf{x}$ but the magnitude is small. The output of the similarity measure will be close to 1, making our approach softly quantifying redundancies.

Truncated soft sign dissimilarity (TSSD), is designed to incorporate the original implementation (Yadav et al., 2023) of the 'redundancy removing' stage. To compute this, we zero-out the parameters of $\mathbf{x}$ and $\mathbf{y}$ if their original values are among the top-k smallest in $\mathbf{x}$ and $\mathbf{y}$ respectively, then apply SSD to the new inputs. After truncation, the redundant values might be set to zero and will make its similarity with the other value to be one. Therefore, this approach can be viewed as a hard approach to quantify redundancy. Note that the truncation introduces many zeros to both $\mathbf{x}$ and $\mathbf{y}$, and we simply do not include the similarity of two zero values in computation because merging two zeros does not affect the merging results and thus should not affect the similarity either. In practice, we set k to 50% of the total parameters of vision weights, thus zeroing half of the parameters.

| Metrics | $K_{seed}$ | | | | Corr. |
|---------|------|------|------|------|-------|
|  | 0 | 50k | 100k | 150k |  |
| L2 | 580.6 | 496.8 | 388.5 | 284.6 | 0.881 |
| Cosine | 0.371 | 0.264 | 0.163 | 0.098 | 0.938 |
| SSD | 0.198 | 0.161 | 0.125 | 0.104 | 0.939 |
| TSSD | 0.066 | 0.037 | 0.015 | 0.006 | **0.971** |
| *Performance drop on COCO retrieval* | | | | | |
|  | 9.16 | 4.50 | 2.80 | 2.60 | |

Table 5: We apply four metrics to modality-specific models trained on different lengths of seed pre-training. The bottom two rows indicate the performance drop on COCO retrieval after model merging, and the right-most column is the Pearson correlation between these metrics and the performance drop. The result shows that TSSD can best reflect the merging outcomes.

**Results.** We apply the aforementioned metrics to modality-specific models using different seed and VL pre-training iterations and show the results along with the performance drop after merging on COCO in Table 5. We observe that the distance indeed relates to the performance drop as their trends are aligned. To investigate what metric best reflects the merging outcomes, we calculate the Pearson correlation (Freedman et al., 2007) between the performance drop and each distance metric. The results in Table 5 reveal that the L2 distance is the least correlated to the performance drop, in comparison to other measures, implying that magnitude may not be a crucial factor for merging. On the other hand, **TSSD shows the highest correlation among all metrics, suggesting a better metric to predict whether two models are mergeable.**

# 8 Conclusion

We explore applying the model merging technique to transform modality-specific models into modality-agnostic models. With comprehensive experiments, we analyze the key factors impacting the merging results, including initialization, merging mechanisms, and model architectures. With seed pre-training and merging methods as simple as interpolation, we improve the merging results of the modality-specific model significantly (+3% on VQA, +7% on COCO image-text retrieval, +25% on NLVR$^2$, +14% on Flickr30k and +3% on ADE20k). Moreover, we demonstrate the merged weight maintains the performance on unimodal tasks, and the proposed approach generalizes to another backbone model. We also discuss an initial recommended recipe for multimodal merging.

## 9 Broader Impact and Limitations

This paper demonstrates the potential to merge a modality-specific VL model to obtain a universal architecture that can handle multiple modalities simultaneously. The findings in this paper address several questions that might be beneficial for future research. Firstly, what representation do the vision and language transformers learn so that the merging performance is better? This helps to understand the fundamental problem of when we can learn a universal model. Secondly, can we merge two transformers that are initialized from unimodal pre-trained weights? This is useful as initializing the models from a related domain would have performance gain. Lastly, can we apply the merging on the models fine-tuned on downstream VL tasks, so as to construct a multi-task modality-agnostic model? We hope our study can inspire future work in this direction.

Though we show seed pre-training is crucial for merging, it shares similar cost issues with other pre-training experiments, leading to high carbon emission and electricity usage that may cause a certain degree of environmental damage and energy cost.

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

## A  Additonal Experimental Details

**Datasets split.** In training, we use the `Karpathy-train` split (Karpathy and Fei-Fei, 2017) for VQA and COCO datasets and use the train set for other datasets. For evaluation, we use `test-dev` split for VQA, test-P split for NLVR$^2$, the validation set for ImageNet and ADE20k, `Karpathy-val` split for COCO and test set for Flickr30k.

**Fine-tuning of ImageNet-1k and ADE20k.** We use the source codes of BEIT (`https://github.com/microsoft/unilm/tree/master/beit`) with our trained checkpoints for fine-tuning.

**Hyper-parameter for fine-tuning.** We show the hyper-parameters used in fine-tuning in Table 6.

| Dataset | Image Size | Learning Rate | Batch Size | Training Epochs |
|---|---|---|---|---|
| VQA | $480 \times 480$ | $3 \times 10^{-5}$ | 128 | 10 |
| COCO | $384 \times 384$ | $6.25 \times 10^{-6}$ | 640 | 20 |
| NLVR$^2$ | $384 \times 384$ | $5 \times 10^{-5}$ | 128 | 10 |
| Flickr30k | $384 \times 384$ | $6.25 \times 10^{-7}$ | 128 | 40 |

Table 6: Hyper-parameters used for fine-tuning.

**Computational Usage.** We use 48xV100 GPUs for seed pre-training and VL pre-training and it takes around 3.5 days for 200k iterations. We use 32xV100 GPUs for VQA and COCO fine-tuning and they take around 7 hours to finish, while we use 16xV100 GPUs for NLVR$^2$ and Flickr30k and they take around 4 hours to finish.

**Number of Parameters.** We display the number of parameters for BEiT$_{base}$ architectures used in this work in Table 7.

| | Modality-specific architecture | Custom Attn | Custom FFN | Custom LN | Modality-agnostic architecture |
|---|---|---|---|---|---|
| Number of parameters | 217M | 151M | 184M | 118M | 118M |

Table 7: Number of parameters for each architecture based on BEiT$_{base}$ when being applied on VQA.

## B  Licenses

- Conceptual Captions: Other

- COCO: Creative Commons Attribution 4.0 License

- Visual Genome: Creative Commons Attribution 4.0 License

- VQAv2: Creative Commons Attribution 4.0 License

- annotations of NLVR$^2$: CC-BY-4.0 (There are no licenses for images as the original authors also do not hold the copyright.)

- Flickr30k: Other

- ADE20k: Creative Commons BSD-3 License Agreement