# OpenReview forum: "An Empirical Study of Multimodal Model Merging"
_EMNLP/2023/Conference — EMNLP 2023 Findings_

### Official Review · Reviewer_Ms1f · 2023-08-03

**Paper Topic And Main Contributions:** 1. This work empirically investigates…
**Soundness:** 3

**Excitement:**

3: Ambivalent: It has merits (e.g., it reports state-of-the-art results, the idea is nice), but there are key weaknesses (e.g., it describes incremental work), and it can significantly benefit from another round of revision. However, I won't object to accepting it if my co-reviewers champion it.

**Reasons To Accept:**

1. This work empirically find multiple interesting observations about multimodal margining, as summarized well in section 1.

2. This work merges well-trained modality-specific models into a modality-agnostic model that can achieve similar performance to the one that is pre-trained from scratch.

3. It proposes two simple metrics that assess the distance between weights to be merged and can serve as an indicator of the merging outcomes.


**Reasons To Reject:**

1. The novelty of this work is limited in that it is an empirical study.

2. The target tasks are rather old: VQA (Goyal et al., 2016) and COCO retrieval (Lin 317 et al., 2014). It would be more interesting with VL tasks with more recent benchmarks.

3. Only a single baseline ViT-B/16 is tested.

4. The proposed two metrics are largely based on those of Yadav 506 et al. (2023).

5. As depicted in Fig.4, the merging results of the modality-specific model increase by +3% on VQA and +7% on COCO image-text retrieval. First, it should justify that these gaps are significant enough, by comparing with other SOTA methods. Second, this is not achieved by any novel idea of this work but by searching for combinations of known simple techniques over key decision factors.



**Reproducibility:**

5: Could easily reproduce the results.

**Reviewer Confidence:**

3: Pretty sure, but there's a chance I missed something. Although I have a good feel for this area in general, I did not carefully check the paper's details, e.g., the math, experimental design, or novelty.

---

> ### Author Rebuttal · Authors · 2023-08-29
>
> Thank you for the thoughtful reviews. We are encouraged that the reviewers found our paper **interesting** [KA3D, v68G, Ms1f], **original** [v68G], and **insightful** [KA3D, v68G]. We are also glad to receive reviewers’ positive comments on our **empirical results** [v68G, Ms1f] and the **analysis of the distance explanation** [v68G, Ms1f]. In the following, we hope our responses can address your concerns, and we will incorporate the updates in the next version.
>
> > *The novelty of this work is limited in that it is an empirical study.*
>
> Thanks for the feedback. However, we respectfully disagree that an empirical study points to a limited novelty, especially at an "empirical" conference (EMNLP). While our paper focuses on empirical findings, we also highlighted the **inherent challenges (the weights are not located in the same basin) and technical variations (model architectures and merging methods)** of combining weights from diverse domains in multi-modal models. Therefore, our study presents **a novel utilization of previous approaches to a novel important application** and can have more contribution given the emergence of large multi-modal models.
>
> > *The target tasks are rather old: VQA (Goyal et al., 2016) and COCO retrieval (Lin 317 et al., 2014). It would be more interesting with VL tasks with more recent benchmarks.*
>
> Our paper focuses on studying how to merge multi-modal models and the downstream performance changes of the chosen models after merging. Therefore, the goal is not to study how the model merging affects the performance of the recent VL benchmarks. VLMo was the state-of-the-art modality-specific model when our project started (still performs in the top 3 on the VQA benchmark), and VQA and COCO are the main tasks VLMo is evaluated on (they are the **most popular and standard tasks adopted in the literature** [1, 2, 3]), so we chose VLMo as the backbone model and VQA and COCO as the target datasets.
>
> That being said, we have also **expanded our study and conducted two more VL experiments on the popular NLVR2 and Flickr30k datasets and two vision experiments on ImageNet and ADE20k**. Due to the short period of the rebuttal phase, we didn’t apply the intermediate fine-tuning (used in the VLMo paper) for Flickr30k, ImageNet-1k, and ADE20k, and only trained ADE20k for 80K steps. We compare our merging approach (seed pre-training + interpolation with a ratio of 0.75) with the other three methods, the modality-specific model, the modality-agnostic method, and the naive merging method. We found that our approach can **boost the merging results compared to naive merging by a large margin** in most tasks and can closely match the modality-agnostic baseline.
>
> | Method             | NLVR2 (Acc) | Flickr30k (Avg IR and TR) | ImageNet-1k (Top-1 Acc) |  ADE20k (mIOU)  |
> |--------------------|-------|-----------|-------------| -------- |
> | Modality-Specific  | 79.71 | 91.24     | 83.25       |  48.27   |
> | Modality-Agnostic  | 78.39 | 88.84     | 82.78       |  46.88   |
> | Naive Merging      | 52.55 | 73.13     | 82.90       |  44.94   |
> | Our Merging        | 77.54 | 87.09     | 83.00       |  48.34   |
>
>
>
> [1] Li, Junnan, Dongxu Li, Silvio Savarese and Steven C. H. Hoi. “BLIP-2: Bootstrapping Language-Image Pre-training with Frozen Image Encoders and Large Language Models.” ArXiv abs/2301.12597 (2023)
>
> [2] Tsimpoukelli, Maria, Jacob Menick, Serkan Cabi, S. M. Ali Eslami, Oriol Vinyals, Felix Hill and Zacharias Janssen. “Multimodal Few-Shot Learning with Frozen Language Models.” Neural Information Processing Systems (2021).
>
> [3] Wang, Wenhui, Hangbo Bao, Li Dong, Johan Bjorck, Zhiliang Peng, Qiang Liu, Kriti Aggarwal, Owais Khan Mohammed, Saksham Singhal, Subhojit Som and Furu Wei. “Image as a Foreign Language: BEiT Pretraining for All Vision and Vision-Language Tasks.” ArXiv abs/2208.10442 (2022)
>
>
> > *Only a single baseline ViT-B/16 is tested.*
>
> As we explained in the previous reply, we focus on how the merging affects the downstream performance of the state-of-the-art modality-specific models, VLMo. While VLMo has two different backbones, ViT-B/16 and ViT-L/16, we chose ViT-B/16 due to resource constraints.
>
> However, as you requested, we conducted **additional experiments on ViT-Tiny/16**. Note that this backbone is not trained by the BEIT unsupervised objective but is trained by the supervised classification objective. Due to the time limit, we only pre-trained the model for 100k steps instead of 200k steps in the paper. For our merging approach, we use 50k steps for both seed pre-training and VL pre-training. We show the results in the following table and **find our technique still performs much better than the naive merging**. The larger gap between our approach and the modality-agnostic baseline with ViT-Tiny/16, compared with the ViT-base model, has been similarly observed in the previous work [4, 5], where they show larger (wider) models are easier to merge.
>
>
> | Method on ViT-Tiny | VQA (Acc)   | COCO (Avg IR and TR)  | NLVR (Acc)  | Flickr (Avg IR and TR) |
> |--------------------|-------|-------|-------|--------|
> | Modality-Specific  | 65.36 | 65.87 | 68.42 | 75.44  |
> | Modality-Agnostic  | 62.25 | 57.63 | 64.9  | 67.38  |
> | Naive Merging      | 55.52 | 47.21 | 51.07 | 16.86  |
> | Our Merging        | 60.02 | 58.11 | 52.89* | 63.96  |
>
> *we found using interpolation ratio of 0.5 can boost the performance to 60.94 and cut the gap to the modality-agnostic model significantly.
>
>
> [4] Ainsworth, Samuel K., Jonathan Hayase and Siddhartha S. Srinivasa. “Git Re-Basin: Merging Models modulo Permutation Symmetries.” ArXiv abs/2209.04836 (2022)
>
> [5] Singh, Sidak Pal and Martin Jaggi. “Model Fusion via Optimal Transport.” ArXiv abs/1910.05653 (2019)
>
> > *The proposed two metrics are largely based on those of Yadav et al. (2023).*
>
> Note that the goal and utilization of the proposed SSD and TSSD metrics are different from the approach of Yadav et al. (2023), and the **definitions of the metrics are also original from our paper**. Yadav studies how to resolve the sign conflicts in the merging while we design the metrics to predict the performance drop of the two models prior to merging (to save the time of training) (lines 574 - 575). Our results show that the **TSSD has the highest correlation with the performance drop**, suggesting that it is the **best indicator to predict the merging outcome**.
>
>
> > *As depicted in Fig.4, the merging results of the modality-specific model increase by +3% on VQA and +7% on COCO image-text retrieval. First, it should justify that these gaps are significant enough, by comparing with other SOTA methods. Second, this is not achieved by any novel idea of this work but by searching for combinations of known simple techniques over key decision factors.*
>
> For the first question: note that our goal is to match the performance of modality-agnostic models via merging, and it can be viewed as one of the upper bounds in the model merging literature [4,5,6]. **Our approach matches this upper bound** (VQA: 73.91 (ours) vs. 73.94, COCO: 78.06 (ours) vs. 78.07) at the end, so we consider our improvement to be significant. Furthermore, to the best of our knowledge, our study **is the first to explore model merging** in VL domains (also pointed out by reviewer v68G), thus there are no existing works (or other SOTA methods) for us to benchmark against. Instead, our main focus is to evaluate various factors that affect the merging outcomes in our VL setting, such as SOTA merging methods (e.g., Interpolation, Modality arithmetic, RegMean), different architectures, and seed pre-training.
>
> For the second question, please refer to our reply to your first question.
>
> [6] Wang, Hongyi, Mikhail Yurochkin, Yuekai Sun, Dimitris Papailiopoulos and Yasaman Khazaeni. “Federated Learning with Matched Averaging.” ArXiv abs/2002.06440 (2020)

---

### Official Review · Reviewer_v68G · 2023-08-05

**Soundness:** 4

**Excitement:**

3: Ambivalent: It has merits (e.g., it reports state-of-the-art results, the idea is nice), but there are key weaknesses (e.g., it describes incremental work), and it can significantly benefit from another round of revision. However, I won't object to accepting it if my co-reviewers champion it.

**Paper Topic And Main Contributions:**

The paper mostly presents an experimental study on weight merging for transformer architectures trained on specific modalities - how it performs multi-modally by merging the weights of language specific arch and vision specific arch. The main contribution is to systematically experiment across 1) effect of initialization 2) merging mechanism and 3) model architecture, to understand how the merged model performs under different combination of these conditions. It arrives at an optimal setup suggestion and shows good performance result.

**Questions For The Authors:**

* It's probably worth clarifying the difference between seed pretraining and VL-pretraining. It took me a while to understand the difference (seed pretraining use shared weights for vision and language inputs while VL-pretraining uses separate weights for each type of input? It's confusing partially because the name "VL-pretraining" indicates modality agnostic training).
* It might be a good idea to dig deeper into the analysis part on why merging works at all and when it works better. Having an explanation-guided merging mechanism would definitely strengthen the paper.

**Reasons To Accept:**

* The problem on weight merging is interesting, and indeed this seems to be one of the first work to study modality-wise weight merging.
* Experiments are thorough and the analysis on the distance explanation provides some insights into why/how merging works.

**Reasons To Reject:**

* Technical-wise seems incremental. Would suggest to study and discuss more on how modality merging differs from previous work on single/multi-task merging. Except the setup difference, are there unique technical challenges or new findings?
* The result is reasonable, but is not good enough to justify the additional steps (seed pretrain -> pretrain -> merging) needed for this procedure, comparing to just do a model agnostic training. Also, it introduces additional hyperparameters (merging ratios etc).

**Reproducibility:**

4: Could mostly reproduce the results, but there may be some variation because of sample variance or minor variations in their interpretation of the protocol or method.

**Reviewer Confidence:**

4: Quite sure. I tried to check the important points carefully. It's unlikely, though conceivable, that I missed something that should affect my ratings.

---

> ### Author Rebuttal · Authors · 2023-08-29
>
> Thank you for the thoughtful reviews. We are encouraged that the reviewers found our paper **interesting** [KA3D, v68G, Ms1f], **original** [v68G], and **insightful** [KA3D, v68G]. We are also glad to receive reviewers’ positive comments on our **empirical results** [v68G, Ms1f] and the **analysis of the distance explanation** [v68G, Ms1f]. In the following, we hope our responses can address your concerns, and we will incorporate the updates in the next version.
>
> > *Technical-wise seems incremental. Would suggest to study and discuss more on how modality merging differs from previous work on single/multi-task merging. Except the setup difference, are there unique technical challenges or new findings?*
>
> In our research on merging models for a novel VL setup, we highlighted the **inherent challenges of combining weights from diverse domains in multi-modal models**. This challenge was shown when we observed suboptimal performance with naive merging attempts, as showcased in Table 1. However, a breakthrough was achieved when **utilizing an optimized initial weight derived from seed pre-training**. This led to merged models performing on par with the modality-agnostic models trained from scratch, as demonstrated in Figure 3, effectively closing the performance discrepancy.
>
> Several unique insights emerged from our study:
>
> * Our complete comparison of various merging techniques revealed that **RegMean performed robustly to hyperparameter choices** (lines 434 - 435). Additionally, using a slightly **higher interpolation ratio in the vision transformer boosted its performance** (lines 438 - 442).
> * While applying the model merging on shared-weight architectures (specialized to multi-modal setups), **we observed that architectures without any shared components outperformed their counterparts** (lines 466 - 470).
>
>
> > *The result is reasonable, but is not good enough to justify the additional steps (seed pretrain -> pretrain -> merging) needed for this procedure, comparing to just do a model agnostic training. Also, it introduces additional hyperparameters (merging ratios etc).*
>
> In this paper, we aim to fairly assess the impact of the factors on merging so we don’t assume the weights exist and train the modality-specific and modality-agnostic models from scratch. That is the reason why we need the three steps (seed pretrain -> pretrain -> merging). However, the optimal case for merging is that we can merge the existing models that are already available online (Huggingface hub), so the process will be efficient as we can directly obtain a modality-agnostic model by **only doing the final step to merge an existing pre-trained modality-specific model**. Moreover, as we introduce in our guidelines (Section 6.4), we have also found that using interpolation ratios of 0.5 and 0.75 is good enough for most experiments (Table 2(a)), or that using RegMean is robust to hyperparameters (Table 3(c)). Providing these insights is also one of the values of this paper.
>
> > *It's probably worth clarifying the difference between seed pretraining and VL-pretraining. It took me a while to understand the difference (seed pretraining use shared weights for vision and language inputs while VL-pretraining uses separate weights for each type of input? It's confusing partially because the name "VL-pretraining" indicates modality agnostic training).*
>
> Thanks for the feedback.  In our paper, **VL pre-training can be conducted on different model architectures, including both modality-specific models and modality-agnostic models**. For example, our experiments include 200k steps of VL pre-training on modality-specific models and modality-agnostic models (the modality-agnostic baseline in Figure 3). The seed pre-training, as you mentioned, is a way to create shared weights for two modalities, and improve the merging results by a large margin. To be more clear, **seed pre-training and VL pre-training are the same for modality-agnostic models, and seed pre-training for a modality-specific model is exactly VL pre-training (or seed pre-training) on a modality-agnostic model**. We use different terms for seed pre-training and VL pre-training to distantiate the different training stages for training the modality-specific model. We understand that the terms can create some confusion, and we will define them more clearly in the next version.
>
>
>
> > *It might be a good idea to dig deeper into the analysis part on why merging works at all and when it works better. Having an explanation-guided merging mechanism would definitely strengthen the paper.*
>
> The intuition and hypothesis of why merging works are discussed in previous works [1, 2], and the main takeaway is that the models become mergeable when they are located in the same basin (basin means a local minima that has low loss everywhere). While those papers proposed the theory, their setup is somewhat **limited to the same modality or even the same tasks**. Our paper digs deeper into studying **how to apply model merging to multi-modalities and when it works better** in this scenario, as well as an analysis in Section 7 to **predict whether two models are mergeable by computing the distance between them in weight space**. Finally, we also release a **guideline on how to merge a multi-modal model** (Section 6.4). We will add the discussion of the position of our work (compared to previous theoretical ones) to the next version.
>
> [1] Ainsworth, Samuel K., Jonathan Hayase and Siddhartha S. Srinivasa. “Git Re-Basin: Merging Models modulo Permutation Symmetries.” ArXiv abs/2209.04836 (2022)
>
> [2] Garipov, T., Pavel Izmailov, Dmitrii Podoprikhin, Dmitry P. Vetrov and Andrew Gordon Wilson. “Loss Surfaces, Mode Connectivity, and Fast Ensembling of DNNs.” ArXiv abs/1802.10026 (2018)
>
> ---
> #### ***We also conduct new experiments on another backbone, ViT-Tiny, to respond to the request of the reviewer Ms1f and new experiments on NLVR2, Flickr30k, ImageNet-1k, and ADE20k to respond to reviewer KA3D. We found the results are quite encouraging. Please refer to our replies to the reviewer Ms1f and KA3D for more information.***

---

### Official Review · Reviewer_KA3D · 2023-08-05

**Typos Grammar Style And Presentation Improvements:** Not found.
**Soundness:** 3

**Excitement:**

3: Ambivalent: It has merits (e.g., it reports state-of-the-art results, the idea is nice), but there are key weaknesses (e.g., it describes incremental work), and it can significantly benefit from another round of revision. However, I won't object to accepting it if my co-reviewers champion it.

**Missing References:**

(Not all, just what I remember:)
1. CoCa: Contrastive Captioners are Image-Text Foundation Models
2. ONE-PEACE: Exploring One General Representation Model Toward Unlimited Modalities
3. SimVLM: Simple Visual Language Model Pretraining with Weak Supervision
4. Unifying Vision-and-Language Tasks via Text Generation
I recommend the authors to have a deeper paper study on multi-modal pretraining.

**Paper Topic And Main Contributions:**

This paper explores the optimal empirical setting (seed pretraining iterations and weight sharing strategy) for multi-modal pretraining and a merging approach to convert modality-specific models into modality-agnostic models. The hyper-param about these settings and different ways of merging are compared on VQA and MSCOCO tasks.

**Questions For The Authors:**

No.

**Reasons To Accept:**

1. Empirical results about seed pretraining iterations and weight sharing strategy may provide insights to obtain better performance for multimodal pretrained models.
2. The research question of model merging in multimodal is an interesting topic which can help to reduce parameters and develop unification models.

**Reasons To Reject:**

I think the most serious problem of this work is the evaluation benchmark only includes VQA and COCO. Just considering multi-modal benchmarks, VQA and COCO are not sufficient enough (when we talk about something on multi-modal pretraining). Moreover, as this work discuss about merging image and text transformers (or say transformer encoder layers) into multi-modal ones, the pre-training, fine-tuning and evaluation must include text-only and image-only tasks. When we only evaluate (limited) multi-modal tasks but talk about "multi-modal model merging" of image & text encoders, the value of this research work will be quite limited. The most referred work VLMo in this paper is also focusing on both multi-modal and visual-only tasks. As a new strategy implemented based on VLMo, at least the same visual-only tasks (ImageNet and ADE20K) should be considered to evaluate whether the merged transformer is actually a "modality-agnostic" one, rather than a module becoming incapable to do single-modal tasks after merging.

**Reproducibility:**

4: Could mostly reproduce the results, but there may be some variation because of sample variance or minor variations in their interpretation of the protocol or method.

**Reviewer Confidence:**

5: Positive that my evaluation is correct. I read the paper very carefully and I am very familiar with related work.

---

> ### Author Rebuttal · Authors · 2023-08-29
>
> Thank you for the thoughtful reviews. We are encouraged that the reviewers found our paper **interesting** [KA3D, v68G, Ms1f], **original** [v68G], and **insightful** [KA3D, v68G]. We are also glad to receive reviewers’ positive comments on our **empirical results** [v68G, Ms1f] and the **analysis of the distance explanation** [v68G, Ms1f]. In the following, we hope our responses can address your concerns, and we will incorporate the updates in the next version.
>
> > *I think the most serious problem of this work is that the evaluation benchmark only includes VQA and COCO. Just considering multi-modal benchmarks, VQA and COCO are not sufficient enough (when we talk about something on multi-modal pretraining). …… As a new strategy implemented based on VLMo, at least the same visual-only tasks (ImageNet and ADE20K) …*
>
> Thank you for the feedback. We conducted experiments mainly on VQA and COCO because they are the **most popular and standard tasks adopted in the literature** [1, 2, 3] (also used in VLMo). However, we agree that more evaluation would make the paper stronger, so we also provide **additional experiments on NLVR2 and Flickr30k** as well as **ImageNet-1k and ADE20k** to cover more VL and vision tasks. Due to the short period of the rebuttal phase, we didn’t apply the intermediate fine-tuning (used in the VLMo paper) for Flickr30k, ImageNet-1k, and ADE20k, and only trained ADE20k for 80K steps.
>
> We compare our merging approach (seed pre-training + interpolation with a ratio of 0.75) with the other three methods, the modality-specific model, the modality-agnostic method, and the naive merging method, and show the results in the table below. We found that **our approach performs much better than the naive merging on NLVR2 and Flickr30k**. However, we found the gap on NLVR2 and Flickr30k between our approach and the modality-agnostic model is a bit larger than the gap on VQA and COCO, where we match the results closely (Figure 4). Since images of VQA and COCO are used in the pre-trained tasks, we hypothesize that the merging results would be better if the domain shift between pre-trained and downstream tasks is smaller.
>
>
> For the ImageNet and ADE20k results, we found that **our merging approach can match or even outperform the baselines**, and these results show that the merging does not destroy the representative power of the single modality. The investigation of studying the different merging outcomes on VL, vision, and language tasks would be an interesting future work.
>
>
> | Method             | NLVR2 (Acc) | Flickr30k (Avg IR and TR) | ImageNet-1k (Top-1 Acc) |  ADE20k (mIOU)  |
> |--------------------|-------|-----------|-------------| -------- |
> | Modality-Specific  | 79.71 | 91.24     | 83.25       |  48.27   |
> | Modality-Agnostic  | 78.39 | 88.84     | 82.78       |  46.88   |
> | Naive Merging      | 52.55 | 73.13     | 82.90       |  44.94   |
> | Our Merging        | 77.54 | 87.09     | 83.00       |  48.34   |
>
> [1] Li, Junnan, Dongxu Li, Silvio Savarese and Steven C. H. Hoi. “BLIP-2: Bootstrapping Language-Image Pre-training with Frozen Image Encoders and Large Language Models.” ArXiv abs/2301.12597 (2023)
>
> [2] Tsimpoukelli, Maria, Jacob Menick, Serkan Cabi, S. M. Ali Eslami, Oriol Vinyals, Felix Hill and Zacharias Janssen. “Multimodal Few-Shot Learning with Frozen Language Models.” Neural Information Processing Systems (2021).
>
> [3] Wang, Wenhui, Hangbo Bao, Li Dong, Johan Bjorck, Zhiliang Peng, Qiang Liu, Kriti Aggarwal, Owais Khan Mohammed, Saksham Singhal, Subhojit Som and Furu Wei. “Image as a Foreign Language: BEiT Pretraining for All Vision and Vision-Language Tasks.” ArXiv abs/2208.10442 (2022)
>
> > *Missing References*
>
> Thank you for providing the reference. We will cite them as well as other VL representation learning papers in the next version.
>
> ---
> #### ***We also conduct new experiments on another backbone, ViT-Tiny, to respond to the request of the reviewer Ms1f. We found the results are quite encouraging. Please refer to our reply to the reviewer Ms1f for more information.***

---

### Meta-Review · Area_Chair_p5Qj · 2023-09-14

**Recommendation:** 3

**Metareview:**

This paper provides a comprehensive empirical study on multimodal model merging for transformer architectures. Three reviewers provided reviews for this paper and these reviews were in consensus. The reviewers appreciated the paper and found the empirical analysis to be interesting. One major feedback by the reviewers was to increase the number of datasets and tasks for evaluation. The authors responded well to this feedback by providing several new experiments in line with previous work that had already been cited. I really appreciate this diligence and effort put in by the authors. The new experiments solidify the findings and increase the impact of this study. With these improvements and clarifications addressed by the authors, reviewers found the work to be technically sound and complete.

---

### Decision · Program_Chairs · 2023-10-07

**Decision:**

Accept-Findings

**Comment:**

This paper provides a comprehensive empirical study on multimodal model merging for transformer architectures. Three reviewers provided reviews for this paper and these reviews were in consensus. The reviewers appreciated the paper and found the empirical analysis to be interesting. One major feedback by the reviewers was to increase the number of datasets and tasks for evaluation. The authors responded well to this feedback by providing several new experiments in line with previous work that had already been cited. I really appreciate this diligence and effort put in by the authors. The new experiments solidify the findings and increase the impact of this study. With these improvements and clarifications addressed by the authors, reviewers found the work to be technically sound and complete.